# Which determinants should be considered to reduce social inequalities in paediatric dental care access? A cross-sectional study in France

**Thomas Marquillier**[1,2]*, **Thomas Trentesaux**[1], **Adeline Pierache**[3,4], **Caroline Delfosse**[1], **Pierre Lombrail**[2], **Sylvie Azogui-Levy**[2,5]

**1** Odontologie pédiatrique, Université de Lille, CHU Lille, Lille, France, **2** Laboratoire Éducations et Pratiques de Santé, LEPS UR 3412, Université Sorbonne Paris Nord, Bobigny, France, **3** ULR 2694—METRICS: évaluation des technologies de santé et des pratiques médicales, Université de Lille, CHU Lille, Lille, France, **4** Département de Biostatistiques, CHU Lille, Lille, France, **5** Faculté d'odontologie, Département de Santé Publique, Université de Paris, Paris, France

* thomas.marquillier@univ-lille.fr

**Data Availability Statement:** The authors proceeded to a deposition within data repository in

## Abstract

Better access to dental care through systemic and educational strategies is needed to lessen the burden of disease due to severe early caries. Our study aims to describe family characteristics associated with severe early caries: parental knowledge, attitudes, practices in oral health and socio-demographic factors. For this cross-sectional study, 102 parents of children aged under 6 years with severe early caries and attending paediatric dentistry service in France completed a questionnaire during face-to-face interviews. Caries were diagnosed clinically by calibrated investigators, using the American Academy of Paediatric Dentistry criteria, and dental status was recorded using the decayed, missing, and filled teeth index. The majority of children were from underprivileged backgrounds and had poor oral health status, with a median dmft index of 10. Parents highlighted the difficulty of finding suitable dental care in private practices. Parents appeared to have good oral health knowledge and engaged in adapted behaviours but showed a low sense of self-efficacy. They perceived the severity of early caries as important but the susceptibility of their child as moderate. The study affirmed the importance of improving the accessibility of paediatric dental care and developing educational strategies to enhance the knowledge, skills, and oral health practices of families.

## Introduction

Severe early childhood caries (S-ECC) is defined as any sign of smooth-surface caries in a child younger than 3 years; and from ages 3 through 5, one or more cavitated, missing due to caries, or filled smooth surfaces in primary maxillary anterior teeth; or a decayed, missing, or filled score of greater than or equal to four (age 3), greater than or equal to five (age 4), or greater than or equal to six (age 5) [1]. The prevalence of S-ECC varies, depending on countries and studies, but it has been estimated between 21% and 41.2% [2–4]. It increases steadily with the

ZENODO (indexed in OpenAIRE). DOI: 10.5281/zenodo.4948448.

**Funding:** The authors received no specific funding for this work.

**Competing interests:** The authors have declared that no competing interests exist.

child's age: 17% when the child is one year old, 36% when 2, 43% when 3, 55% when 4, and 63% when 5 [5]. The disease has individual consequences for children, their family, and community and is considered a major public health problem worldwide [6]. It mainly affects children from underprivileged backgrounds [7].

Patient management usually consists of dental care, under nitrous oxide sedation or general anaesthesia, performed by paediatric dentists. In France, paediatric dental care is insufficient and unevenly distributed [8–10]. In addition, the demand for care exceeds the availability, causing significant waiting times that, in turn, worsen children's health condition. In disadvantaged groups characterized by low socio-economic status, low literacy level (i.e. a person's capacity to obtain, process, and understand basic health information and services needed to make appropriate health decisions [11]), and less frequent use of care, S-ECC is highly recurrent and can become a chronic disease [12–14]. Prevention strategies have led to a decrease in the dmft (decayed-missing-filled tooth) index [15,16], but since they do not take into account barriers to dental care (e.g. literacy, oral health knowledge) or dental access (e.g. numerical, geographical, and financial accessibility, especially for patients with partial or no social coverage), they are not sufficient to reach the most affected population groups. The current strategies contribute to increasing social inequalities in oral health [6].

Levesque et al. proposed a conceptual framework for healthcare access (Fig 1) describing determinants of demand and supply [17]. In this model, the knowledge, attitudes, and practices in oral health and the socio-demographic characteristics of families are determinants of access to paediatric dental care. There has been no study exploring these determinants in France. Our study aims to describe the knowledge, attitudes, practices, and socio-demographic characteristics of families of children with severe early caries to identify strategies for improving access to care and reducing social inequalities in oral health.

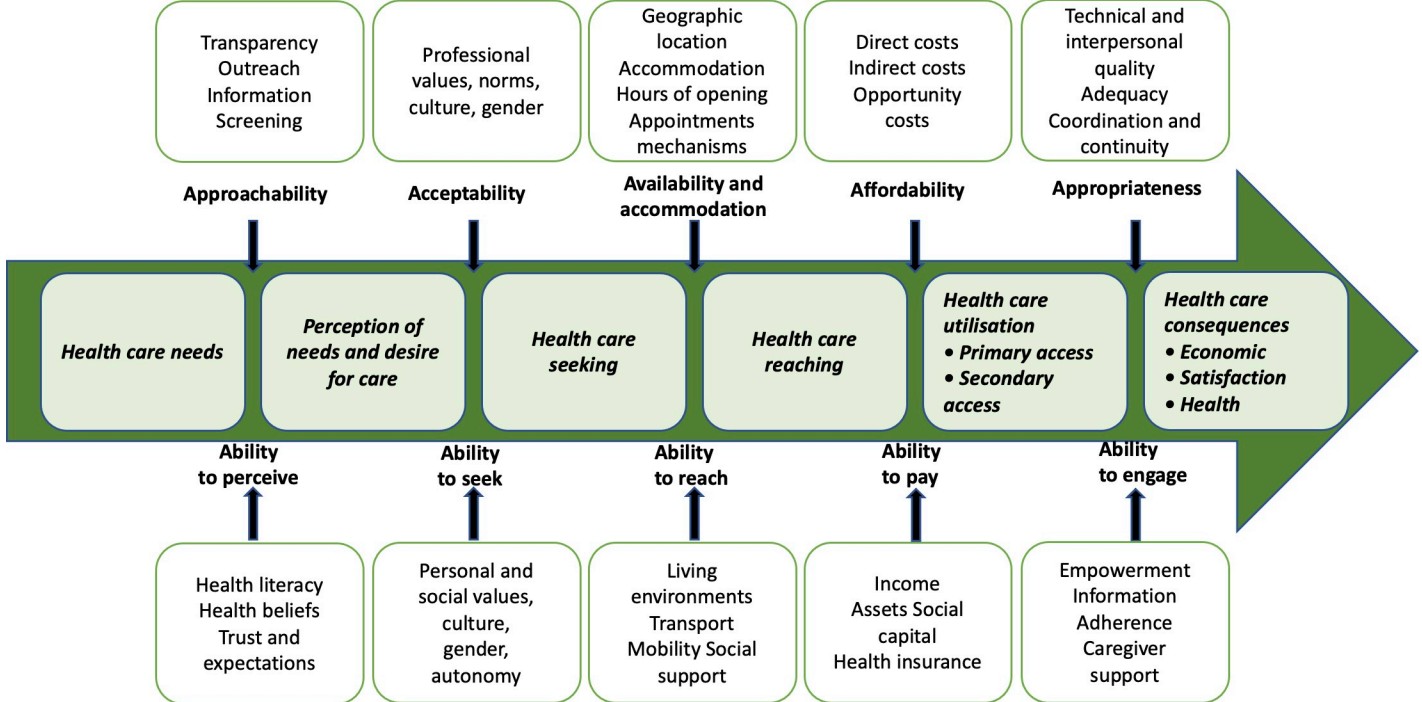

**Fig 1. A conceptual framework of access to health care.** Levesque proposed a model of the determinants of access to health care [17].

## Methods

### Study design, population, and ethics

This cross-sectional study with a prospective recruitment was conducted following the Strengthening the Reporting of Observational studies in Epidemiology (STROBE) statements. A total of 102 parents of children with severe early caries attending the paediatric dentistry department of Lille University Hospital for treatment participated in the study. The sample represents the Haut-de-France region. The French Personal Protection Committee approved the study, and the protocol was recorded on ClinicalTrials.gov (Identifier: NCT04195607). A Participant Information Sheet was provided, and a written consent from all participants was obtained prior to their inclusion in the study.

Participants were selected according criteria, between November 2019 and October 2020 at Lille University Hospital (North of France), Odontology Service, Pole of Medical and Surgical Specialties, Functional Unit of Paediatric Dentistry. Each child patient must be affiliated with France's social security insurance. Children who are not affiliated with social security insurance, have a person taking care of them daily who is not their mother or father, and who do fulfill the following conditions will not be included in the study: (i) they have a serious medical condition (such as leukemia . . .), and/or (ii) their parents do not speak French. Each patient can be included only once in the study (they may be included in the study at either the point of initial treatment or relapse, but not both). Families selected for the study will be questioned in the presence of the patient, on the day of their consultation, and in one of the hospital's consulting rooms. Patients who fulfil the inclusion criteria and are willing to consult will be selected for study until the required number of subjects is met. Children will be selected after they have had a clinical examination validating a diagnosis of severe early caries. A child can be included in the study at any point in their course of treatment, including before, during, or after treatment. The study consists of two steps: (1) clinical examination of the child and (2) interview of the parent.

### Diagnosis and clinical examination of the child

The clinical examination is a usual procedure. Children were examined by the principal investigator or by calibrated examiners (inter-rater reliability k = 0.89; mean intra-rater reliability k = 0.97) in the presence of one of their parents. Examiners used an examination tray consisting of a mirror, a probe, tweezers, and pads. According to the definition of the American Academy of Paediatric Dentistry (AAPD), children with S-ECC were included. The dmft index was used to record caries for each primary tooth present [18]. The teeth were examined under the operating light after drying them with a pad to determine the dmft index.

### Parent interview

Data on family predictors were collected using a structured questionnaire developed by the researcher (TM) based on the literature [19]. The questionnaire was written in French and explored two types of variables: quantitative and qualitative. The questionnaire was piloted in a non-target sample of three parents of children with S-ECC. The formulation of certain questions were later modified.

The participants completed the questionnaire during a face-to-face interview with the investigator who performed the clinical examination. Interviews were conducted in a consulting room of the hospital.

The questionnaire consisted of three sections with a total of 34 items. The first section concerns the child's medical history (i.e. presence of disease or use of medication) and oral health

practices (i.e. frequency of brushing and its supervision by an adult, frequency of food intake, consumption of sweet foods, drinks consumed during meals and during the day) and the parent's dental history (i.e. dental health, dental treatment) and tobacco use. The second section is about the family's socio-demographic characteristics (i.e. child's age and country of birth; mother and father's age, country of birth, educational level, last employment, occupational status, and marital status; number of people living at home; number of children in the family; siblings; social benefits). The investigator then obtained their health insurance information and asked the reason for consultation, the avoidance of dental care (for the parent and for the child), and the importance of the cost of dental care. In the third section, we explored parents' oral health knowledge and attitudes (i.e. locus of control, sense of self-efficacy, perceived importance of oral health-related behaviours, and characteristics of oral health belief model) [19]. Finally, we determined parental oral health literacy level through a single question which is "How often do you need someone to help you when reading instructions, leaflets, or other written documents" [20].

## Statistical analysis

Continuous variables were expressed as means (±standard deviation, SD) in the case of normal distribution or medians (interquartile range, IQR) otherwise. Normality of distribution was assessed using histograms and the Shapiro-Wilk test. Categorical variables were expressed as numbers (percentage). Association of dmft index with the presence of an associated disease was tested using Mann-Whitney U test and the association of dmft index with child's age was tested using Spearman's rank correlation coefficient. Comparisons in child's characteristics according with the frequency of a tooth brushing at least twice a day were performed using Chi-square tests (or Fisher' exact tests when expected cell frequency was <5) except for child's age where a Mann-Whitney U test was used.

Statistical testing was done at the two-tailed α level of 0.05. Data were analyzed using the SAS software package, release 9.4 (SAS Institute, Cary, NC, USA).

## Results

### Socio-demographic characteristics of the family and parental oral health literacy level

Table 1 show the socio-demographic characteristics of the study participants. Of the 102 children, 57 (55.9%) were boys. The majority of children (95.1%, N = 97) were born in France. Mean age was 4.0 years (± 1.1). Most children (77.5%, N = 79) were accompanied by their mother, while 19.6% (N = 20) were accompanied by their father. The mothers' mean age weas 33.5 years (± 6.6). Among mothers, 71.6% (N = 73) were born in France and 74.5% (N = 76) had a level of education less than or equivalent to a bachelor's degree. Regarding their last employment, 43.1% of mothers were employees (N = 44), while 40.2% had no profession (N = 41). Stay-at-home mothers comprised 54.6% (N = 56), whereas 37.3% of mothers were employed (N = 38). In contrast, the fathers' mean age was 37.1 years (± 8.0). The majority of fathers (66.3%, N = 67) were born in France, and 76.3% (N = 77) had a level of education less than or equivalent to a bachelor's degree. Regarding their last employment, 40.6% were blue-collar workers (N = 41), and 30.7% were employees (N = 31). Most fathers (83.2%, N = 84) were engaged in professional activity. Among parents, 79.4% (N = 81) were married or in a similar relationship, while 16.7% (N = 17) were divorced or separated. Regarding the number of children in the family, 34.3% (N = 35) of families have two children (including half-brother and half-sister), 30.4% (N = 31) have three children, 39.2% (N = 40) have four children, while

**Table 1. Socio-demographic characteristics of the family, health status and oral health practices of the child.**

| | Variable | Category | N = 102[a] | |
|---|---|---|---|---|
| **Socio-demographic characteristics of the child** | Child gender, Males | | 57 (55,9) | |
| | Child country of birth | France | 97 (95,1) | |
| | | Outside the France | 5 (4,9) | |
| | Child age | <2 years | 3 (2,9) | |
| | | <3 years | 11 (10,8) | |
| | | <4 years | 12 (11,8) | |
| | | <5 years | 34 (33,3) | |
| | | <6 years | 42 (41,2) | |
| | | Mean age (± SD) | 4,0 (±1,1) | |
| | Accompanying person | Father | 20 (19,6) | |
| | | Mother | 79 (77,5) | |
| | | Other | 3 (2,9) | |
| | Parents marital status | Married or civil partnership or cohabiting | 81 (79,4) | |
| | | Divorced or separated | 17 (16,7) | |
| | | Single | 4 (3,9) | |
| | Number of children | 1 | 15 (14,7) | |
| | | 2 | 35 (34,3) | |
| | | 3 | 31 (30,4) | |
| | | > 3 | 21 (20,6) | |
| | Place in the siblings | 1st | 36 (35,3) | |
| | | 2nd | 24 (23,5) | |
| | | 3rd | 23 (22,6) | |
| | | 4th and beyond | 19 (18,6) | |
| | Number of people living at home | 2 | 6 (5,9) | |
| | | 3 | 14 (13,7) | |
| | | 4 | 40 (39,2) | |
| | | 5 | 24 (23,5) | |
| | | 6 and more | 18 (17,7) | |
| **Socio-demographic characteristics of the parents** | | | Mother (N = 102)[a] | Father (N = 101)[a] |
| | Country of birth | France | 73 (71,6) | 67 (66,3) |
| | | Outside the France | 29 (28,4) | 34 (33,7) |
| | Educational level | No diploma | 10 (9,8) | 10 (9,9) |
| | | Certificate of general education | 14 (13,7) | 14 (13,8) |
| | | Certificate of professional competence or equivalent | 20 (19,6) | 31 (30,7) |
| | | Bachelor's degree or equivalent | 32 (31,4) | 22 (21,8) |
| | | 2 years after bachelor | 11 (10,8) | 11 (10,9) |
| | | Higher diploma | 15 (14,7) | 13 (12,9) |
| | Last employment | Farmer | 0 (0) | 0 (0) |
| | | Craftsman, shopkeeper or entrepreneur | 1 (1,0) | 8 (7,9) |
| | | Senior executive or higher intellectual profession | 7 (6,9) | 8 (7,9) |
| | | Intermediate profession | 6 (5,9) | 4 (4,0) |
| | | Employee | 44 (43,1) | 31 (30,7) |
| | | Worker | 3 (2,9) | 41 (40,6) |
| | | Without profession | 41 (40,2) | 9 (8,9) |
| | Occupational status | Employed | 38 (37,2) | 84 (83,1) |
| | | Student or apprenticeship | 1 (1,0) | 0 (0) |
| | | Unemployed | 5 (4,9) | 11 (10,9) |
| | | Retired | 0 (0) | 2 (2,0) |
| | | Stay-at-home parent | 56 (54,9) | 0 (0) |
| | | Another situation | 2 (2,0) | 4 (4,0) |
| | Social benefits | | 54 (52,9) | |

*(Continued)*

**Table 1.** (Continued)

| | Variable | Category | N = 102[a] | |
|---|---|---|---|---|
| **Dental status, medical history, and oral health practices of the child** | | | | N = 102[a] |
| | Smooth surfaces | Decayed | 87 (85,3) | |
| | Primary maxillary anterior teeth attacked | Decayed, filled or missing (caries) | 92 (90,2) | |
| | Disease | Long-term disease | 12 (11,8) | |
| | | Asthma | 6 (5,9) | |
| | | Gastroesophageal reflux | 2 (2,0) | |
| | Tooth brushing | At least once a day | 86 (84,3) | |
| | | Morning before breakfast | 13 (15,1) | |
| | | Morning after breakfast | 61 (71,8) | |
| | | Noon | 11 (12,9) | |
| | | Evening before dinner | 3 (3,5) | |
| | | Evening after dinner | 79 (91,9) | |
| | | Supervised oral hygiene | 68 (66,7) | |
| | Frequency of food intake | > 4 per day | 75 (73,5) | |
| | | Daily sweet foods | 94 (92,2) | |
| | | Sweet foods once a day | 25 (26,7) | |
| | Sweet food | Sweet foods twice a day | 38 (40,4) | |
| | | Sweet foods three times | 18 (19,1) | |
| | | Sweet foods four times a day or more | 13 (13,8) | |
| | Main drinks during meals | Tap water | 6 (5,9) | |
| | | Bottled water | 72 (71,3) | |
| | | Soda | 4 (4,0) | |
| | | Fruit juice | 8 (7,9) | |
| | | Other | Fruit syrup | 10 (9,9) |
| | | | Milk | 1 (1) |
| | Sweet drinks | Daily sweet drinks: | 64 (62,7) | |
| | | -Sweet drinks once a day | 25 (39,1) | |
| | | -Sweet drinks twice a day | 20 (31,2) | |
| | | -Sweet drinks three times | 5 (7,8) | |
| | | -Sweet drinks four times a day or more | 14 (21,9) | |

[a]Values are expressed as numbers (percentage).

23.5% (N = 24) have five children staying in the same home. Among children with severe early caries, 35.3% (N = 36) were the first child, 23.5% (N = 24) were the second child, and 22.5% (N = 23) the third child in the family. Notably, 54% of the families received social aid (i.e. children and housing). To determine parental oral health literacy level, the parents were asked the question, "How often do you need someone to help you when reading instructions, leaflets, or other written documents from your doctor or pharmacy?", to which 56.9% (N = 58) answered never, 16.7% (N = 17) almost never, 20.6% (N = 21) sometimes, 2.9% (N = 3) often, and 2.9% (N = 3) always.

These results provide in the first part the socio-demographic characteristics and concern the gender, the country of birth and the age of the child with severe early caries. They also specify the relationship with the carer, the marital status of the parents, the number of children in the family, the place of the child with severe early caries in the siblings and finally the number

of people living in the same household. The second part describe the socio-demographic characteristics of the parents and concern their country of birth, their education level, their last employment, their occupational status and their perception of social benefits. The last part provide informations firstly on severe early caries (number of smooth surfaces decayed, primary maxillary anterior teeth attacked), secondly on the child medical history (in particular: long-term disease, asthma or gastroesophageal reflux). Then, these results expose oral health practices: frequency of tooth brushing, frequency of consumption of sweet foods, sweet drinks and the main drinks consumed during meals.

## Dental status, medical history, and oral health practices

Table 1 shows the children's dental status, medical history, and oral health practices (e.g. brushing and feeding). Notably, 11.8% of the children (N = 12) had a pathology, and 7.8% (N = 8) had long-term treatment. Of the 102 children median dmft index was 10 (interquartile range 8 to 13). Analysis of the dmft index according to the child's age gives us a median dmft index of 8 at 1 year, 10 at 3 years, and 11 at 5 years. A positive significant correlation was found between child's age and dmft index (r = 0.23, p = 0.019). Dmft index was not significantly associated with the presence of an associated disease (median with pathology 9 (IQR, 8 to 10.5), without 11 (IQR, 8 to 14), p = 0.22).

Among parents, 84.3% (N = 86) reported brushing the teeth of their child at least once a day, with 71.8% (N = 61) doing so in the morning after breakfast, and 91.9% (N = 79) doing so in the evening before bedtime. 93.3% (N = 42) of girls have at least one tooth brushing compared to 77.2% (N = 44) of boys (p = 0.026). Children with at least one tooth brushing (N = 86) are on median 4 years old, as those without tooth brushing (N = 16) (p = 0.23). Most parents (66.7%, N = 68) supervised their child's tooth brushing. Regarding food intake, 73.5% of children (N = 75) eat more than four times a day, and 92.2% (N = 94) eat sweet food (e.g. pastries, chocolate bars, etc.) daily. While 77.3% of parents (N = 78) indicated that their child drinks water during mealtime, 62.7% (N = 64) reported that their child consumes sugary drinks daily. 83% of children who consume sweet foods on a daily basis have at least one tooth brushing compared to 100% for those who do not consume (p = 0.35). 80% of children who eat more than 4 times a day have at least one tooth brushing compared to 96.3% who do not eat more than 4 times a day (p = 0.063).

## Oral health knowledge, attitudes and behavior

Parental oral health knowledge is summarized in Table 2. Only 20% of the parents interviewed appeared to have poor oral health knowledge, but majority of them do not know about fluoride or its role in dental care. Parental oral health self-efficacy which is the belief that the parent has in his or her ability to perform a task is summarized in Table 3. Notably, 31% of parents had a positive sense of self-efficacy, that is, they strongly agree or agree with the implementation of behaviours adapted to their child's oral health. Parental oral health behaviours are presented in Table 4 where 81% of parents agree with good oral health behaviours. Most parents (64.7%, N = 66) thought that they are responsible for the presence of early childhood caries (i.e. internal locus of control), while 35.3% (N = 36) thought that the occurrence of the disease was not under their control (i.e. external locus of control). The majority of parents (71%) considered engaging in favourable behaviours to promote their child's oral health (e.g. checking their child's mouth regularly, brushing twice a day with fluoride toothpaste, going to the dentist regularly, avoiding sweet foods and drinks even at night) of high importance. Characteristics concerning the health belief model are summarized in Table 5. Perceived susceptibility was tempered; in fact, 53% of parents thought that most children have dental caries. The severity of

**Table 2. Parental oral health knowledge.**

| Statement | True[a] | False[a] | I don't know[a] |
|---|---|---|---|
| A child can brush his teeth alone at 4 | 44 (43,1) | 58 (56,9) | 0 |
| A child needs to have his first dental visit at 6 | 28 (27,5) | 74 (72,5) | 0 |
| Child teeth need to be brushed once a day | 14 (13,7) | 88 (86,3) | 0 |
| Temporary teeth are not important | 5 (4,9) | 93 (91,2) | 4 (3,9) |
| There is no need to go to the dentist unless the child has a problem | 10 (9,8) | 90 (88,2) | 2 (2,0) |
| Fluoride toothpaste is better to brush child teeth | 53 (52,0) | 32 (31,35) | 17 (16,65) |
| Bacteria cause caries | 82 (80,4) | 16 (15,7) | 4 (3,9) |

| Statement | Good[a] | Bad[a] | Not good or bad[a] | I don't know[a] |
|---|---|---|---|---|
| Eating after brushing teeth and before going to bed | 2 (1,95) | 98 (96,1) | 2 (1,95) | 0 |
| Eating crisps | 5 (4,9) | 82 (80,4) | 15 (14,7) | 0 |
| Drinking sodas | 1 (1,0) | 97 (95,1) | 4 (3,9) | 0 |
| Sharing a toothbrush with your child | 0 | 102 (100) | 0 | 0 |
| Using the same spoon to taste the food and feed the child | 3 (2,9) | 80 (78,45) | 19 (18,65) | 0 |
| Protecting child's teeth with fluoride | 43 (42,2) | 18 (17,6) | 17 (16,7) | 24 (23,5) |
| Looking in your child's mouth every month to see any changes | 95 (93,1) | 2 (2,0) | 4 (3,9) | 1 (1,0) |

[a]Values are expressed as numbers (percentage).

These results provide informations about the parental oral health knowledge through two sets of questions. Parents were asked to answer true, false or I don't know to the first 7 questions on basic knowledge. To the next 7 questions on knowledge of oral health behaviours, they answered good, bad, neither good nor bad or I don't know.

the disease was perceived as important for the child's health, with 93,1% of parents agreeing with the statement "Dental problems can be serious for a child". Barriers to oral health behaviours were perceived as low, but 55.9% of parents thought that "It is difficult to prevent [their]

**Table 3. Parental oral health self-efficacy.**

| Statement | Strongly disagree[a] | Disagree[a] | Neutral[a] | Agree[a] | Strongly agree[a] | I don't know[a] |
|---|---|---|---|---|---|---|
| You check your child's teeth and gum carefully every month | 6 (5,9) | 5 (4,9) | 22 (21,6) | 23 (22,5) | 46 (45,1) | 0 |
| You regularly take your child the dentist for check-up | 6 (5,9) | 8 (7,8) | 15 (14,7) | 18 (17,6) | 55 (53,9) | 0 |
| You use fluoridated toothpaste for your child | 12 (11,8) | 7 (6,9) | 15 (14,7) | 9 (8,8) | 41 (40,2) | 18 (17,6) |
| Your child doesn't take anything except water after brushing his teeth and before going to sleep | 22 (21,6) | 8 (7,8) | 7 (6,9) | 5 (4,9) | 60 (58,8) | 0 |
| You prevent your child from frequently eating sweets | 15 (14,7) | 8 (7,8) | 15 (14,7) | 19 (18,6) | 45 (44,1) | 0 |
| You prevent your child from putting something that has been in someone else's mouth into their own | 5 (4,9) | 4 (3,9) | 5 (4,9) | 12 (11,8) | 76 (74,5) | 0 |
| Fluoride varnish has already been applied to the teeth of your child | 66 (64,7) | 4 (3,9) | 4 (3,9) | 3 (2,9) | 9 (8,8) | 16 (15,7) |
| You prevent your child from drinking sodas | 16 (15,7) | 12 (11,8) | 11 (10,8) | 22 (21,6) | 41 (40,2) | 0 |
| You avoid putting your child to bed with a sweet bottle | 12 (11,8) | 5 (4,9) | 3 (2,9) | 9 (8,8) | 72 (70,6) | 1 (1,0) |
| Teeth of your child are brushed twice a day | 17 (16,7) | 8 (7,8) | 11 (10,8) | 15 (14,7) | 51 (50,0) | 0 |

[a]Values are expressed as numbers (percentage).

These results provide informations on parental self-efficacy in oral health through a set of 10 questions whose answers are based on a Likert scale.

**Table 4. Parental oral health behaviors.**

| Statement | Useless[a] | Useful[a] | I don't know[a] |
|---|---|---|---|
| Take your child for a dentist for check-up or cleaning | 2 (2,0) | 100 (98,0) | 0 |
| Take the child for his first visit before a year | 75 (73,5) | 26 (25,5) | 1 (1,0) |
| Brush the child's teeth twice a day or more | 4 (3,9) | 98 (96,1) | 0 |
| Brush your teeth twice a day or more | 6 (5,9) | 96 (94,1) | 0 |
| Help children brush their teeth when they are under 6 | 4 (3,9) | 98 (96,1) | 0 |
| Eat sweets less than once a day | 20 (19,6) | 82 (80,4) | 0 |
| Consume sugary drinks less than once a day | 20 (19,6) | 82 (80,4) | 0 |
| Use fluoride toothpaste for the child | 24 (23,5) | 60 (58,8) | 18 (17,6) |
| Do not eat or drink (anything other than water) after brushing teeth and before going bed | 5 (4,9) | 97 (95,1) | 0 |

[a]Values are expressed as numbers (percentage).

These results provide informations on parental oral health behaviors through a set of 9 questions to which parents answer useful, useless or I don't know.

child from eating or drinking sweet foods". Nevertheless, the parents believed that the perceived benefits of positive oral health behaviours are important.

## Paediatric dental care and parent's dental history

Characteristics related to paediatric dental care are shown in Table 6. Of the 102 families, 40.2% (N = 41) lived more than 30 minutes away from the service provider that takes care of their child. Most parents (79.4%, N = 81) previously consulted with dentists in private practice but were not satisfied with the care their child received; in fact, 35.8% (N = 29) had visited two or more dentists. Among parents, 43.1% (N = 44) reported that it is difficult to find suitable dental care for their young child. Notably, 64.7% (N = 66) of children were directly referred by their private dentist, 8% (N = 8) were referred by their doctor, while 23% went to the hospital spontaneously with their parent or upon the advice of a friend, a paediatrician, or a family doctor. Concerning the avoidance of care, 26.5% of parents (N = 27) already renounced dental care for themselves, with 33.3% (N = 9) giving the high cost of care as the main reason. Twenty-eight parents (27.5%) indicated that the cost of dental care is the main barrier to getting treatment for themselves, while 11 (10.8%) reported the same for their child. The majority of parents (93.1%, N = 95) stated that they never gave up dental care for their child, with 95.1% (N = 97) admitting that they have no difficulty paying for medication or health services for their family. Regarding health insurance, 51% (N = 52) had a universal health insurance and a private supplementary insurance, 43.1% (N = 44) had a universal health insurance and a solidarity supplementary insurance (i.e. French social benefits providing access to care, reimbursement of care, or medicines to any person residing in France who is not already covered by another compulsory health coverage), while 3,9% (N = 4) had a universal health insurance only without supplementary health insurance.

More than half of parents (52.9%, N = 54) considered their dental condition as good, 12.7% (N = 13) very good, 27.5% (N = 28) bad, and 6.9% (N = 7) very bad. Notably, 90.2% (N = 92) indicated that they already have a dental problem (e.g. tooth decay, gum disease), 94.1%

**Table 5. Parental oral health belief model.**

| Statement | Strongly disagree | Disagree | Neutral | Agree | Strongly agree | I don't know |
|---|---|---|---|---|---|---|
| Most of children have dental caries | 9 (8,8) | 15 (14,7) | 22 (21,6) | 26 (25,5) | 28 (27,5) | 2 (2,0) |
| Your child will have caries in the next few years | 15 (14,7) | 31 (30,4) | 18 (17,6) | 29 (28,4) | 8 (7,8) | 1 (1,0) |
| My child can have a carie as soon as his first tooth has erupted | 34 (33,3) | 16 (15,7) | 14 (13,7) | 16 (15,7) | 20 (19,6) | 2 (2,0) |
| It is unlikely that my child will have problems with his teeth | 21 (20,6) | 32 (31,4) | 20 (19,6) | 20 (19,6) | 9 (7,8) | 1 (1,0) |
| Dental problems can be serious for a child | 0 | 1 (1,0) | 5 (4,9) | 10 (9,8) | 85 (83,3) | 1 (1,0) |
| Having bad teeth affect child's daily life | 61 (59,8) | 27 (26,5) | 2 (2,0) | 4 (3,9) | 8 (7,8) | 0 |
| Dental problems are not as important as other health problems | 74 (72,5) | 19 (18,6) | 3 (2,9) | 5 (4,9) | 1 (1,0) | 0 |
| It is difficult to take my child for the dentist for regular check-up | 46 (45,1) | 19 (18,6) | 7 (6,9) | 17 (16,7) | 13 (12,7) | 0 |
| It is difficult to prevent my child from eating or drinking sweet foods | 14 (13,7) | 20 (19,6) | 11 (10,8) | 21 (20,6) | 34 (33,3) | 2 (2,0) |
| I don't have any problem making sure my child's teeth are brushed before he goes to sleep | 15 (14,7) | 12 (11,8) | 7 (6,9) | 19 (18,6) | 47 (46,1) | 2 (2,0) |
| It is a problem for my child to have fluoride varnish on his teeth | 27 (26,5) | 18 (17,6) | 14 (13,7) | 1 (1,0) | 4 (3,9) | 38 (37,3) |
| I don't have any problem making sure my child's teeth are brushed with fluoride toothpaste twice a day | 7 (6,9) | 12 (11,8) | 16 (15,7) | 19 (18,6) | 33 (32,4) | 15 (14,7) |
| It is unlikely that my child will have caries if their teeth are brushed with fluoride toothpaste twice a day | 15 (14,7) | 28 (27,5) | 12 (11,8) | 25 (24,5) | 11 (10,8) | 11 (10,8) |
| It is unlikely that my child will have caries if he goes to the dentist for regular check-up | 15 (14,7) | 18 (17,6) | 12 (11,8) | 33 (32,4) | 23 (22,5) | 1 (1,0) |
| It is unlikely that my child will have caries if I stop him from eating lots of sweet foods | 8 (7,8) | 16 (15,7) | 17 (16,7) | 28 (27,5) | 33 (32,4) | 0 |
| It is unlikely that my child will have caries if an adult helps him brush their teeth until he is 6 | 12 (11,8) | 21 (20,6) | 12 (11,8) | 32 (31,4) | 25 (24,5) | 0 |
| It is unlikely that my child will have caries if the dentist puts fluoride varnish on his teeth | 8 (7,8) | 12 (11,8) | 26 (25,5) | 11 (10,8) | 6 (5,9) | 39 (38,2) |

[a]Values are expressed as numbers (percentage).

These results focus on the oral health belief model which is studied through a set of 17 questions where parents are asked, on a Likert scale, to what extent they agree or disagree with the following statements.

(N = 96) have had dental care, 23.5% (N = 24) have already used a dental emergency service for themselves, and 42.2% (N = 43) have had a dental abscess. Regarding their last visit to a dentist, 39.2% (N = 40) went to a dentist in the last six months, 29.4% (N = 30) in the last year, 14.7% (N = 15) more than a year ago, and 16.7% (N = 17) more than three years ago. Among parents, 34.3% (N = 35) reported smoking.

## Discussion

In this cross-sectional study, we aimed to describe the determinants to paediatric dental care access, following Levesque et al.'s model, as represented by structural determinants (e.g. socio-demographic characteristics, availability of dental care) and individual determinants (e.g. parents' oral health knowledge, attitudes, and practices). There has been no previous study that explored all these determinants in France. Among children, attitudes and health-related practices are established through primary socialization, while health-related behaviours are

**Table 6. The characteristics of the use of pediatric dental care.**

| Variable | Category | N = 102[a] | |
|---|---|---|---|
| **Distance from home to Hospital (minutes)** | < 10 | 16 (15,7) | |
| | < 20 | 28 (27,5) | |
| | < 30 | 17 (16,7) | |
| | < 40 | 19 (18,6) | |
| | < 50 | 9 (8,8) | |
| | < 60 | 6 (5,85) | |
| | > 60 | 7 (6,85) | |
| **Number of dentists previously consulted (concerning people who consulted in private)** | 1 | 52 (64,15) | |
| | 2 | 17 (20,95) | |
| | 3 and more | 12 (14,9) | |
| **Person motivating the visit** | Pediatrician or family doctor | 8 (8,0) | |
| | Private dentist | 66 (64,7) | |
| | The parent himself/a friend | 23 (23,0) | |
| | | | |
| **Reasons for avoidance of care (concerning people who renounces care)** | | Parents (N = 27) | Child (N = 7) |
| | Cost of care | 9 (33,3) | 3 (42,9) |
| | Anxiety | 8 (29,6) | 2 (28,6) |
| | Care consideration | 7 (25,9) | 1 (14,25) |
| | Patient's refusal | 2 (7,5) | 0 |
| | Transports | 1 (3,7) | 1 (14,25) |

[a]Values are expressed as numbers (percentage).

These results concern the use of paediatric dental care: The distance between the place of care and the home, the number of dentists previously consulted, the person who motivated the visit, the reasons for having renounced care previously (for the child or the parent).

adopted and learned from caregivers [21]. To promote children's oral health, caregivers need to have suitable knowledge, acquire specific skills, and establish health-oriented practices [22]. Parents also need to have a sufficient level of literacy for them to navigate the health system.

## S-ECC as a marker of inequalities

Firstly, we highlighted findings regarding S-ECC. The average age of children in this study was 4 years old, which is the same as in Tinanoff's study [5]. This can be linked to the fact that children between 0 and 3 years old with severe early caries are difficult to care for and that the majority of parents bring their children to the dentist when they are between 4 and 5 years old. Concerning gender, our results align with Peltzer's study–boys are more affected by caries than girls (i.e. 57 versus 45) [23]. The mean dmft score (10.4 ± 4.0) in our study is higher than other studies on S-ECC (9.1 ± 3.35 in Romania [24], 8.17 ± 2.94 in China [4], and 1.01 ± 2.37 in India [3]). The dfmt index increases with the child's age, which is in line with Tinanoff's study. Among children, 11.8% had an associated pathology and 7.8% had long-term treatment. Although these values are lower than those in other studies [25,26], long-term medication is a risk factor for developing early caries. According to a retrospective cohort study conducted in Taiwan, children with asthma and receiving medications had higher dental caries prevalence and higher rate of severe caries than children without asthma [27]. In our study, having a long-term disease does not seem to increase the caries score in children. One possible explanation for these results is that in children with long-term disease, even if the risk is higher, parents could have more preventive oral health behaviours to avoid decompensating their pathology. The majority of children (77.5%) in our study were accompanied by their mother–

a finding similar to a study conducted in Korea [28]. This could be explained by the fact that there are more mothers than fathers without a profession. Among the parents we interviewed, 28.4% of mothers and 33.7% of fathers were born outside France. According to Östberg, having a foreign-born parent is one of the main risk factors for dental caries [29]. People of foreign origin consult less because access to care is more difficult, particularly because of the language barrier. In our study and in previous studies, low family socio-economic level was associated with dental caries in children [30]. We also found that the majority of parents have a low level of education, with 74.5% of mothers and 76.3% of fathers having a level of education less than or equivalent to a bachelor's degree. Notably, 56% of mothers stay at home, while 83.2% of fathers are employed, mainly as blue-collar workers, and 54% of parents receive social benefits. These data match with earlier works [4,23,31], affirming that early childhood caries is a marker of social inequalities [7]. To ensure equity in access to oral healthcare, we must strengthen oral health promotion and education at different stages of a child's life and target vulnerable populations. Moreover, it is necessary to address barriers to healthcare access, particularly structural determinants.

## Structural explanations to consider

Secondly, we proposed structural explanations regarding access to paediatric dental care. The parents' care pathway is difficult, and they consult too late because of long waiting times. The diagnosis of S-ECC is not made by the family doctor but by a dentist with whom parents do not consult early enough. After the diagnosis, the recourse to specialized structures is complex. In fact, 79.4% of children were not taken into care in private offices or were not treated in private practice due to the unavailability of a specialized paediatric dentist or the inability of general dentists to treat very young children. In our study, 64.7% of children were directly referred by their private dentist. In France, many dentists do not cover care for people from underprivileged backgrounds who only have solidarity supplementary insurance [32]; hence, they are referred to hospitals where waiting times are long. Delaying care can consequently cause complications for children [33,34]. In our study, 40.2% of families live more than 30 minutes away from the hospital that takes care of their child, and this makes dental check-ups challenging. Difficulties related to access to paediatric dental care emerge as the main barriers explaining severe early caries [2]. For 43.1% of families, it was difficult to find a dentist to care for their child, and this underlines the need for sufficient and efficient care to avoid caries relapse. Concerning the renunciation of care, 26.5% of parents already renounced dental care for themselves, with 33.3% stating the high cost of care as the main reason. For 27.5% of parents, the cost of dental care is a barrier to getting treatment for themselves, while 10.8% reported the same for their child even though only 3,9% reported not having supplementary insurance. It is difficult for these parents to avail of specialized procedures (e.g. nitrous oxide sedation) because these are expensive and often not covered in private offices. Although dental insurance can be considered as a lever for patients to seek care, in France, almost all patients have health insurance (public and private) covering the costs of the main dental treatments. If patients pay the cost of care, other reasons may explain the non-use of dental care, for example, dental anxiety [35]. Overall, this financial barrier, which is linked to supplementary health insurance, is less significant than main obstacle–insufficient health care provision (e.g. specialized paediatric dentist).

## The role of individual components

Finally, the study focused on individual explanations. The majority (80%) of parents have a good knowledge of oral health, but they do not know much about fluoride. Regarding the

statement "It is better to use fluoride toothpaste when brushing children's teeth", only 52% of parents answered "true", while only 42% agreed with the statement "Protecting child's teeth with fluoride". These findings are consistent with those BaniHani et al.'s study [36], and affirms that parents' knowledge remains theoretical and not operational. This should be highlighted because using fluoride toothpaste is one of the most effective ways to prevent early caries in children. The AAPD 2008 recommends that the age of the first consultation should be no later than 1 year; however, in our study, only 27.5% of the parents interviewed thought that this is necessary. This finding is in line with those of previous studies: Higher knowledge does not necessarily translate to greater adherence with recommended oral health behaviours or improved oral health outcomes among children [37]. In our study, only 31% of parents had a positive sense of self-efficacy. As a behavioural determinant, self-efficacy reflects the extent to which a person feels capable of engaging in recommended health behaviours [38]. Self-efficacy is a predictor of maternal oral health behaviours and children's oral health outcomes (e.g. dmft) [39]. This shows that knowledge is a central element, but it must be transformed into skills and actions to promote children's health. Regarding parental health literacy level, more than half of parents stated that they never needed help with reading written documents from their doctor or pharmacy. Nevertheless, it is necessary to consider interventions to empower families and increase their level of literacy to facilitate access to healthcare structures (e.g. understanding how they work and resolving barriers to access). According to Chi, oral health educational interventions can improve self-efficacy [40], but these should be culturally and linguistically tailored [41]. In our study, only 19% of parents do not believe in good oral health behaviours, which is an interesting finding because parents play an important role in the oral health of their children and are the primary decision-makers regarding health and health-related behaviours. Notably, 64.7% of parents reported having an internal locus of control. According to Albino, who conducted a research among American Indians, children who have parents with an internal locus of control have smaller increases in dmft over the course of a prevention program than those whose parents have an external locus of control [42]. The majority of parents (71%) believed that it is important to engage in good oral health behaviours, although this was not enough to implement them since behaviour change is a complex matter. Oral health is conditioned by belief. According to Wilson, mothers with higher oral health knowledge perceive greater benefits from adherence to recommended oral health behaviours and have greater confidence in their ability to manage their children's oral health [39].

Most parents (66%) considered their oral health to be positive (i.e. either good or very good). Children whose parents feel that their health is poor are 3.9 times more likely to develop ECC [43]. In fact, 90.2% of parents in our study indicated that they already have a dental problem. According to Roberts et al., children who have caregivers with tooth loss have significantly greater caries prevalence than those whose caregivers have no tooth loss [44]. Mothers who have high levels of untreated caries are more than three times as likely to have children who have an increasing extent of caries experience [45]. In our study, 14.7% of parents had their last dental consult more than a year ago, while 16.7% had one more than three years ago. According to Bozorgmehr, there is no significant relationship between parents' and children's frequency of dental visits [46]. Notably, many parents generally prioritize their child's health over theirs for whatever reason. In our study, only 23.5% of parents have used a dental emergency service for themselves.

Concerning oral health practices for their chil, 84.3% of parents reported brushing at least once a day; this percentage is higher than that in a study conducted in Taiwan where 61% of children with S-ECC had at least one daily brushing [47]. Although brushing twice a day is recommended, frequency of brushing has not been validated as an indicator of early caries,

according to Nobile et al. [31]. In our study, 66.7% of parents reported supervising their child when toothbrushing. Studies have highlighted the link between lack of brushing supervision and the development of early caries [4,48,49]. Our study is in agreement with the literature which shows that girls generally have more positive oral health behaviours, however the gap remains limited with the boys [50]. According to Murthy, for children under 6 years old, toothbrushing should be performed by parents [51]. In addition to advising children to start toothbrushing at a very early age, parents should be advised to supervise them until they are at least 6 years old. Regarding food intake, 73.5% of children in our study eat more than four times a day, and this is in line with a previous study published in 2013, indicating that children with S-ECC eat more than five times a day (5.26 ± 1.64) [52]. Most children (92.2%) eat sweet foods daily. Daily consumption of sweet foods has been associated with early caries between 1 and 2 years of age, especially when it exceeds 10% of the recommended energy intake [53,54]. While 77.32% of parents indicated that their child consumes water during mealtime, 62.7% stated that their child consumes sugary drinks daily. This is consistent with the literature [55], but the percentage in our study is less than that in a previous study conducted in China where 76 to 82% of children who drink sweetened drinks daily developed early childhood caries [56]. Our study found that children who eat sweet foods more than four times a day brush their teeth more frequently. However, according to a study carried out in a French adolescent population, positive eating behaviours are associated with more frequent brushing [57]. Our results could be explained by the fact that in our very young population, parents who have less control over their children's eating practices would be more involved in brushing their teeth, with the objective of limiting dental caries progression.

## Perspectives

Like any cross-sectional study, our study has limitations. Our small sample consisted of children diagnosed with S-ECC in a regional hospital centre. While the representativeness may be discussed, it is necessary to keep in mind that in France young children with severe early caries are not treated in private practices, thus the Hospital is the first place of recourses for dental care. Nevertheless, the findings cannot be generalised to the entire paediatric population. In addition, we used a questionnaire that may be affected by response bias due to social desirability. Another factor that may cause response bias is that parents who have consulted before in private practices may have received prevention information. It would be interesting to know the origin of parents' oral health knowledge (social media, doctor, dentist. . .).

We conclude that our referral activity in paediatric dental care confirms a public health problem, mainly related to the access to care. We identified two types of determinants: structural (e.g. lack of specialized paediatric dentists in private practices and lack of preventive measures to avoid the disease) and individual (e.g. need to improve parental operational knowledge and skills, attitudes, practices, and literacy). Clinicians can address individual determinants by using strategies, such as therapeutic patient education. However, structural measures are required to address structural determinants and enhance dental care accessibility, ensure primary and tertiary prevention, and improve treatment.

## Acknowledgments

The authors thank the clinical research department of Lille University Hospital for their support.

## Author Contributions

**Conceptualization:** Thomas Marquillier, Pierre Lombrail, Sylvie Azogui-Levy.

**Data curation:** Thomas Marquillier, Adeline Pierache.

**Formal analysis:** Adeline Pierache.

**Investigation:** Thomas Marquillier, Thomas Trentesaux, Caroline Delfosse.

**Methodology:** Thomas Marquillier, Pierre Lombrail, Sylvie Azogui-Levy.

**Project administration:** Thomas Marquillier, Caroline Delfosse.

**Resources:** Thomas Marquillier, Thomas Trentesaux, Caroline Delfosse.

**Supervision:** Caroline Delfosse, Pierre Lombrail, Sylvie Azogui-Levy.

**Validation:** Thomas Marquillier, Adeline Pierache, Sylvie Azogui-Levy.

**Visualization:** Thomas Trentesaux.

**Writing – original draft:** Thomas Marquillier.

**Writing – review & editing:** Thomas Marquillier, Adeline Pierache, Caroline Delfosse, Pierre Lombrail, Sylvie Azogui-Levy.

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
