## [Decision Letter · Decision Letter 0]

6 May 2021

PONE-D-21-12152

Which determinants should be considered to reduce social inequalities in paediatric dental care access?  A cross-sectional study in France.

PLOS ONE

Dear Dr. MARQUILLIER,

Thank you for submitting your manuscript to PLOS ONE. After careful consideration, we feel that it has merit but does not fully meet PLOS ONE’s publication criteria as it currently stands. Therefore, we invite you to submit a revised version of the manuscript that addresses the points raised during the review process.

We look forward to receiving your revised manuscript.

Kind regards,

Frédéric Denis, Ph.D.

Academic Editor

PLOS ONE

Journal Requirements:

Reviewers' comments:

Reviewer's Responses to Questions

**Comments to the Author**

1. Is the manuscript technically sound, and do the data support the conclusions?

Reviewer #1: Yes

Reviewer #2: Partly

2. Has the statistical analysis been performed appropriately and rigorously? 

Reviewer #1: N/A

Reviewer #2: No

3. Have the authors made all data underlying the findings in their manuscript fully available?

Reviewer #1: Yes

Reviewer #2: Yes

4. Is the manuscript presented in an intelligible fashion and written in standard English?

Reviewer #1: Yes

Reviewer #2: Yes

5. Review Comments to the Author

Reviewer #1: The study aims to describe family characteristics associated with severe early caries, based on a clinical evaluation of 102 children whose parents were interviewed. The topic seems original, insofar as it is the first study to our knowledge with such a design, and as the topic is of interest.

Some comments should however be addressed:

Introduction

1) L.55-56 : « The prevalence of S-EEC varies, … between 21% and 41.2% ». Is this regardless to age ? Are there differences in children of 3 or 6 year old ?

2) L.60 : « in France, paediatric dental care is insufficient and unevenly distributed ». Do you have a reference for this statement ?

3) L.65 : « Prevention strategies have led to a decrease in the dmft index ». Do you have a reference for this statement ? Or is it reference 5 ?

Methods

1) L.85 : « the sample represents the Haut-de-France region ». It is unclear how it is representative ? To what extent can you affirm that it is representative ?

2) L.90-97 : Although we understand the global design of the study, I think authors should be clearer in the conduct of the study, precisely (1) children : clinical examination / (2) parents : interview. In the method section, it is sometimes unclear who we are talking about : parent or child ?

3) L.112 : what if both parents accompanied their child ? How did you choose, if you have made a choice ?

4) L.121-122 : « asked the renouncement », « asked the effect of the cost of dental care ». I think these should be rephrased.

5) L.125 : although the « single question » appears in the results section, I think authors should mention this question here in the methods section.

Results

1) L.174 : authors set a threshold at « at least once a day » for the toothbrushing frequency. Why once ? Recommendations are of a twice a day brushing, it would have been relevant to have a « twice a day » line, which could have allowed a comparison with previous national studies (e.g. Fernandez de Grado G. et al. Plos One, 2021).

2) Table 3 : « sweet food > 4 per day : 75 ». Some lines further « sweet foods four times a day or more : 13 ». I think the first line refers to the eating frequency rather than the sweet food intake ?

3) L.195 : there is a unsollicited parenthesis after « health »

4) Table 5 : 1st statement : before going « to » bed

5) In general, and for better clarity, is it possible to bring together some tables ?

Discussion

1) I think there shoud be subheadings in the discussion section, which would lead to a much more comfortabel reading.

2) L.277 : is the term « caregiver » correct ? I think authors are speaking of parents. It is confusing how parents are caregivers, and authors should not lead to any misunderstanding between parents or nurses / dentists / other therapists.

3) L.285 : 8.5±3.82 versus 9.1±3.35 in Romania : why « versus » ? To what is the first value compared ?

4) L.286-287 : « 7.8% had a long-term treatment … is a risk for developing early caries ». Authors seem to have the data to make a comparison between children with or without long-term treatment regarding their dmft index. It would be interesting to have an idea if there is a significant higher dmft index in this study ?

5) L.289 : « 28.4% of mothers and 33.7% of fathers were born in France ». In table 2, these values refer to « born outside of France ». Is there a mistake in line 289 ?

6) L.306 : is the reference 25 accurate ? « In France, many dentists do not cover car for people from underprivileged backgrounds… », but ref. 25 deals with Italy.

7) L.308 : with reference 27, another reference would be relevant regarding the consequences linked to delaying care : North S. et al. J Paediatr Dent, 2007 17 :105-9

8) L.319-321 : considering the results showed at lines 250-255, is there really a financial barrier, or is it rather a musunderstanding or a lack of understanding/knowledge about the healthcare system and its possibilities ?

9) L.354 : again, the word « caregivers » is confusing. Are you speaking of the parents ?

10) L.361 : « 84.3% of parents reported brushing at least once a day » : for themselves or for their children ?

11) L. 368-371 : it would be interesting to established a parallel with these associated factors and the frequency of a toothbrushing at least twice a day, since it is also associated to food intake, perceived family wealth, etc. See Fernandez de Grado G. et al. Plos One, 2021.

12) L.376-379 : « may not be generalizable » : this joins my previous comment (introductiotn section, comment 1). It would have been interesting to compare these results with those of a control group. Is there a reason why no such comparison was conducted ?

13) L.383 : « need to improve parental knowledge ». The study showed that the knowledge would not be that bad. The key point in my opinion is to find a way to upgrade skills/practices in relation to knowledge.

In general, authors should be attentive to the concordance of times. Frequently, past times are mixed with sentences in the present time (l.114, l.147, l.176-179, …). Please have a second reading of the whole manuscript.

Reviewer #2: Dear author,

This study, although interesting, deserves some clarification in the event of publication.

First of all, and it is mentioned in your discussion, this study is not representative of a population, neither geographically nor an age group. Its sample is too limited. Its main selection bias is to be a sample of children referred by the private dentist, probably not all children with severe early caries.

Please specify clearly including or excluding criteria (serious medical conditions?).

Please specify which analytical statistics tests have enabled you to confirm the elements of your discussion, and show the P value for each one.

In socioeconomic factors, have you studied the monthly household budget in example? Or the influence of place of life (city vs country...). Have you been able to determine which level of parental education gives the child the greatest risk? From the mother? from father? both? Is there a correlation? Too much "According to..." in the discussion. Why didn't you use Liskert scale in table 4?

For the oral part, which part of children use a toothbrush zero or one time a day? You wrote "sense of self-efficacy 31%". Do you mean that parents believe in doing the right things for their child?At the end of the consultation, did you proceed to an oral health education session, and if so, did you see the child again to validate the changes in belief and ability?

The analysis of oral health beliefs is very interesting and well done.

In my opinion, the discussion should be revised by specifying that it is a descriptive study on a small sample, not representative of the French population or of the age group studied. I would prefer that analytical statistics come to validate the assertions of the discussion.

Sincerely yours

6. PLOS authors have the option to publish the peer review history of their article (what does this mean?). If published, this will include your full peer review and any attached files.

Reviewer #1: No

Reviewer #2: No

---

## [Author Response · Author response to Decision Letter 0]

18 Jun 2021

A response letter has been uploaded previously. Its contents are given below.

--

Dear Reviewers,

On behalf of all authors, I would like to thank you for the time you spent providing constructive comments on this article.

We have made the corrections, modifications, and revisions that you have requested to improve the manuscript. We also responded to your comments, the details of which you will find in the attached pages. We believe that the study is now much stronger and clearer as a result of your review.

We hope you enjoy reading it.

Yours sincerely, 

Thomas MARQUILLIER

 

General comments

The authors mentioned that the study protocol is currently being published in BMJ Open. The revised manuscript has been submitted

The authors reformatted this manuscript to follow the journal's guidelines.

The authors used a professional service (Scribbr) to check language use in the manuscript.

The authors proceeded to a deposition within data repository in ZENODO (indexed in OpenAIRE).

DOI: 10.5281/zenodo.4948448 

Reviewers’ comments

Comments to the Author

1. Is the manuscript technically sound, and do the data support the conclusions?

Reviewer #1: Yes

Reviewer #2: Partly

2. Has the statistical analysis been performed appropriately and rigorously? 

Reviewer #1: N/A

Reviewer #2: No

The suggested analyses and the tests used have been added.

3. Have the authors made all data underlying the findings in their manuscript fully available?

Reviewer #1: Yes

Reviewer #2: Yes

The authors proceeded to a deposition within data repository in ZENODO (indexed in OpenAIRE).

DOI: 10.5281/zenodo.4948448 

4. Is the manuscript presented in an intelligible fashion and written in standard English?

Reviewer #1: Yes

Reviewer #2: Yes

The authors used a professional service (Scribbr) to check language use in the manuscript.

5. Review Comments to the Author

Reviewer #1: The study aims to describe family characteristics associated with severe early caries, based on a clinical evaluation of 102 children whose parents were interviewed. The topic seems original, insofar as it is the first study to our knowledge with such a design, and as the topic is of interest.

Some comments should, however, be addressed:

Introduction

1) L.55-56: « The prevalence of S-EEC varies, … between 21% and 41.2% ». Is this regardless of age? Are there differences in children of 3 or 6 years old? 

As indicated in the manuscript, the prevalence of S-EEC depends on the country, the studies, and the age of the child. The older the child is, the higher is the prevalence, and the more the pathology is expressed. This is logical since the eruption of teeth follows an order: As the child grows, they develop more teeth that can be affected by the condition.

According to Tinanoff, “Finding from these 72 reports that the mean caries prevalence for 1-year-olds was 17%, and greatly increased to 36% in 2-year-olds. Additionally, the 3-, 4-, and 5-year-olds’ mean caries prevalence were 43%, 55%, and 63%, respectively.” 

The authors have made this clarification in the manuscript.

We found a positive significant correlation between child’s age and dmft index (r = 0.23, p = 0.019). Analysis of the dmft score according to the child's age generated a median dmft score of 8 at 1 year, 10 at 3 years, and 11 at 5 years. These results are in line with the literature. 

2) L.60: « in France, paediatric dental care is insufficient and unevenly distributed ». Do you have a reference for this statement? 

Paediatric dentistry is not an officially recognised speciality in France, so it is difficult to provide an exact distribution. Nevertheless, three references affirm the fact that paediatric dental care is unevenly distributed and is insufficient in number in France.

We added the following references to support our assertion:

Muller-Bolla M, Clauss F, Davit-Béal T, Manière MC, Sixou JL, Vital S. Oral and dental care for children and adolescents in France. Le Chirurgien-Dentiste de France. 2018;1806–1807. 

Dominici G, Muller-Bolla M. Activity of private « paediatric » dentists in France. Rev Francoph Odontol Pediatr. 2017;4(12):152–158. 

Fock-king M, Muller-Bolla M. Analysis of the growing demand for paediatric dentistry treatment in hospitals. Clinic. 2018;39:411–17. 

3) L.65: « Prevention strategies have led to a decrease in the dmft index ». Do you have a reference for this statemen? Or is it reference 5? 

This statement is supported not only by reference 5 but also by the two references below:

Abhishek M. Comprehensive review of caries assessment systems developed over the last decade. RSBO. 2012;9(3):316–21.

Frazao P. Epidemiology of dental caries: When structure and context matter. Braz Oral Res. 2012;26(1):108–14.

Methods

1) L.85: « The sample represents the Haut-de-France region ». It is unclear how it is representative? To what extent can you affirm that it is representative? 

The representativeness of the patient population of a university hospital is indisputable since it is the only place where patients in the region seek care. The provision and the standard of care in private practices are difficult to access and dissuasive for these patients. The study thus aims to improve the quality of the service offered in the region by adding an adapted educational component to improve the knowledge and skills of this specific population. The representativeness seems to be in line with expected practices at a regional level.

The goal is to study a specific group of affected children to illustrate a practical reality, so it is a proxy measure. Severely affected children show recurrence of caries, which necessitate the implementation of an educational programme that addresses the needs of this population.

2) L.90-97: Although we understand the global design of the study, I think authors should be clearer in the conduct of the study, precisely (1) children : clinical examination / (2) parents : interview. 

The authors have clarified this point.

In the methods section, it is sometimes unclear who we are talking about: parent or child? 

The authors have clarified this in the methods section.

3) L.112: What if both parents accompanied their child? How did you choose, if you have made a choice? 

In general, only one parent is allowed to accompany the child in the paediatric care area. It is assumed that the parent accompanying the child is the one who cares for the child the most, knows the child the best, and reassures the child when needed, and is thus the parent who is most willing to answer questions.

4) L.121-122: « asked the renouncement », « asked the effect of the cost of dental care ». I think these should be rephrased. 

The sentence has been reworded.

5) L.125: Although the « single question » appears in the results section, I think authors should mention this question here in the methods section. 

The question has been added in the methods section.

Results

1) L.174: Authors set a threshold at « at least once a day » for the toothbrushing frequency. Why once? Recommendations are twice a day brushing; it would have been relevant to have a « twice a day » line, which could have allowed a comparison with previous national studies (e.g. Fernandez de Grado G. et al. Plos One, 2021). 

This is true; however, in very young children, especially those from low-income backgrounds who are severely affected by tooth decay, oral hygiene is generally absent. We assumed that proper brushing at least once a day with fluoride toothpaste and under parental supervision is already an important first step for these children. The time of brushing (i.e. evening before bedtime) and parental supervision are here studied in preference to the frequency of brushing performed. At the start of the study, the parameter ‘brushing twice a day’ was to be recorded, but after testing the questionnaire, the item was adapted to fit the reality and to avoid overloading the already dense questionnaire.

2) Table 3: « sweet food > 4 per day : 75 ». Some lines further « sweet foods four times a day or more : 13 ». I think the first line refers to the eating frequency rather than the sweet food intake? 

This is correct. Revision has been made: 75% is the frequency of food intake, and 13 is the frequency of sweet food intake.

3) L.195: There is an unsolicited parenthesis after « health » 

This has been corrected.

4) Table 5: 1st statement : before going « to » bed 

This has been corrected.

5) In general, and for better clarity, is it possible to bring together some tables? 

Thank you for your comment. The authors have bring together some tables.

Discussion

1) I think there should be subheadings in the discussion section, which would lead to a much more comfortable reading. 

The authors have added subheadings to the discussion section.

2) L.277: is the term « caregiver » correct ? I think authors are speaking of parents. It is confusing how parents are caregivers, and authors should not lead to any misunderstanding between parents or nurses / dentists / other therapists. 

The term “caregiver” is also used for stepfamilies and single-parent families and is a common formulation in the Anglo-Saxon literature.

3) L.285 : 8.5±3.82 versus 9.1±3.35 in Romania : why « versus » ? To what is the first value compared ? 

This is an error that has been modified.

4) L.286-287: « 7.8% had a long-term treatment … is a risk for developing early caries ». Authors seem to have the data to make a comparison between children with or without long-term treatment regarding their dmft index. It would be interesting to have an idea if there is a significant higher dmft index in this study. 

This is indeed an interesting comparison. In the literature, children with long-term medical conditions that necessitate medication (e.g. asthma) have a higher prevalence of dental caries (Wu FY, Liu JF. Asthma medication increases dental caries among children in Taiwan: An analysis using the National Health Insurance Research Database. J Dent Sci. 2019 Dec;14(4):413-418. doi: 10.1016/j.jds.2019.08.002.). It thus makes sense to add this comparison in the results and discussion sections. 

In our sample, dmft index was not significantly associated with the presence of pathology (median with pathology 9 [IQR, 8 to 10.5], without pathology 11 [IQR, 8 to 14]; p = 0.22).

These results can be explained by the fact that despite increased risk of early caries among children with long-term pathologies, parents might have more preventive oral health behaviours to prevent the pathology.

5) L.289: « 28.4% of mothers and 33.7% of fathers were born in France ». In table 2, these values refer to « born outside of France ». Is there a mistake in line 289?

This is an error that has been modified.

6) L.306: Is reference 25 accurate? « In France, many dentists do not cover car for people from underprivileged backgrounds… », but reference 25 deals with Italy.

This is a mistake. We only kept reference 26.

7) L.308: With reference 27, another reference would be relevant regarding the consequences linked to delaying care : North S. et al. J Paediatr Dent, 2007 17 :105-9 

Thank you for this suggestion. We have added this reference.

8) L.319-321: Considering the results showed at lines 250-255, is there really a financial barrier, or is it rather a misunderstanding or a lack of understanding/knowledge about the healthcare system and its possibilities? 

There is indeed a lack of knowledge of the healthcare system and its possibilities on the one hand; on the other hand, the specific treatment of very young children (e.g. sedation, crowns on baby teeth, etc.) in towns requires fees that are not covered by insurance. These extra fees are common in France. People with universal health insurance and a solidarity supplementary insurance are often labelled in private practice as disadvantaged. They are reluctant to pay fees that are not covered, so there is a clear financial barrier to dental care.

9) L.354 : Again, the word « caregivers » is confusing. Are you speaking of the parents?

The term “caregiver” is also used for stepfamilies and single-parent families and is a common formulation in the Anglo-Saxon literature. It refers, in particular, to parents, but it is important not to be exclusive about the term.

10) L.361: « 84.3% of parents reported brushing at least once a day » : for themselves or for their children? 

This statement has been clarified.

11) L. 368-371: It would be interesting to establish a parallel with these associated factors and the frequency of toothbrushing at least twice a day since it is also associated to food intake, perceived family wealth, etc. See Fernandez de Grado G. et al. Plos One, 2021. 

Thank you for your comment. As suggested, we have added comparisons of child characteristics and frequency of toothbrushing. 

Regarding gender, 93.3% of girls have at least one toothbrushing compared to 77.2% of boys (p = 0.026). Comparing these data with age, children with at least one toothbrushing and those without toothbrushing are, on mean, 4 years old (p = 0.23). 

Regarding the consumption of sweet products, 83% of children who consume sweet foods on a daily basis have at least one toothbrushing compared to 100% of those who do not consume sweet foods (p = 0.35). Meanwhile, 80% of children who eat more than four times a day have at least one toothbrushing compared to 96.3% of those who do not eat more than four times a day (p = 0.063). These results have been included and discussed in the discussion section.

12) L.376-379: « may not be generalizable » : This joins my previous comment (introduction section, comment 1). It would have been interesting to compare these results with those of a control group. Is there a reason why no such comparison was conducted? 

We understand and appreciate your comment and would like to provide some background regarding our thought process prior to the study. We considered conducting a case-control survey but recognised that such would be extremely difficult, as it would require matching the ethnic or social origin variable, for example. This would make it nearly impossible to find controls. In contrast, a descriptive study in a patient population without a control group is appropriate in our case. The representativeness of the patient population of a university hospital has been discussed above. Although there can be no question regarding “representativeness” in the statistical sense of the term, there may be questions at an epidemiological level in terms of Berkson’s bias. Children admitted to the hospital are those with the worst oral health, and it is necessary to identify the causes of their poor oral health to improve response and intervention. The study therefore aims to improve the quality of this service by adding an adapted educational component that addresses the needs of this specific population in the region.

13) L.383: « need to improve parental knowledge ». The study showed that the knowledge would not be that bad. The key point in my opinion is to find a way to upgrade skills/practices in relation to knowledge. 

Thank you for this pertinent observation. We have made it clear that our goal is to find a way to improve the operational knowledge and skills of parents and not only their theoretical knowledge, which is already partly satisfactory.

In general, authors should be attentive to the concordance of times. Frequently, past times are mixed with sentences in the present time (l.114, l.147, l.176-179, …). Please have a second reading of the whole manuscript.

Thank you for your attention. We used a professional service to check language use in the manuscript.

Reviewer #2: 

This study, although interesting, deserves some clarification in the event of publication.

First of all, and it is mentioned in your discussion, this study is not representative of a population neither geographically nor an age group.

We understand your comment and would like to provide some background to our thought process prior to the study. We considered that a case-control survey would be extremely difficult to set up as it would require matching the ethnic or social origin variable, for example. This would make it nearly impossible to find controls. By contrast, a descriptive study in a patient population without a control group—with the biases it includes—is attractive in our case. The representativeness of the patient population of a university hospital has been discussed above.

Its sample is too limited. 

It is true that the sample in this study (N = 102) may seem limited at first sight. Nevertheless, the analysis in this first original study conducted in France on a population of children with early caries is mainly descriptive; therefore, it seems fitting to identify the first results. Cross-sectional studies have strengths and limitations that need to be highlighted beforehand, so you are right to point this out. 

Its main selection bias is its sample of children referred by the private dentist. 

The representativeness of the patient population of a university hospital is indisputable since it is the only place where patients in the region seek care. The provision and the standard of care in private practices are difficult to assess and dissuasive for these patients. Although there can be no question regarding “representativeness” in the statistical sense of the term, there may be questions at an epidemiological level in terms of Berkson’s bias. Children admitted to the hospital are those with the worst oral health, and it is necessary to identify the causes of their poor oral health to improve response and intervention. The study, therefore, aims to improve the quality of this service by adding an adapted educational component that addresses the needs this specific population in the region.

Probably not all children with severe early caries 

Children with early caries are not only referred by private dentists, they also come by themselves. The hospital is the first place of recourse for these young children who are not treated by private dentists in France. The only bias is that we do not know how many children did not manage to reach the hospital.

Please specify clearly inclusion or exclusion criteria (serious medical conditions?). 

This subsection has been clarified.

Please specify which analytical statistics tests have enabled you to confirm the elements of your discussion, and show the P value for each one. 

We described the statistical tests used in the statistical analysis. Association of dmft index with the presence of pathology was tested using Mann-Whitney U test, while the association of dmft index with child’s age was tested using Spearman’s rank correlation coefficient. Comparisons of child characteristics and frequency of toothbrushing were performed using Chi-square tests (or Fisher’s exact tests when the expected cell frequency was <5), except for child’s age for which Mann-Whitney U test was used.

For socioeconomic factors, have you studied the monthly household budget, for example? 

No, this data has not been collected directly, but it can be deduced from the parents' socio-professional category. It is difficult to collect this data in populations that are not very well off, and of what use would it be? In France, income data is difficult to collect because of differences in salaries and allowances and because of a certain degree of reticence when providing this type of information. In general, it is common practice to collect this data via the profession or deduce is from the patient’s level of social protection (e.g. the social minimum).

Or the influence of place of life (city vs country...) 

This data is interesting, but it was not directly studied by choice since the survey was already cumbersome. However, we evaluated the travel time between the patient’s home and the hospital. As the hospital is located in the centre of a metropolis, it is easy to deduce the location of the participants.

Have you been able to determine which level of parental education gives the child the greatest risk? From the mother? From father? Both? Is there a correlation?

This information is not easy to determine in a cross-sectional study. It entails complex information that can be estimated by aggregating several data in a more robust study. The authors will consider this relevant remark for their future work.

Too much "According to..." in the discussion 

We have reworded sentences containing this phrase.

Why didn't you use Likert scale in Table 4? 

This is possible, but our questionnaire was based on the literature, which uses more true or false or do not know for the knowledge part.

For the oral part, which part of children use a toothbrush zero or one time a day? 

As it is indicated that 84% of children brush their teeth at least once a day. From this data, we deduced the percentage of those who do not brush.

You wrote "sense of self-efficacy 31%". Do you mean that parents believe in doing the right things for their child? 

Self-efficacy is the belief that an individual can perform a task. This has been specified in the results section.

At the end of the consultation, did you proceed to an oral health education session, and if so, did you see the child again to validate the changes in belief and ability? 

No, this was not possible during the consultation, and changes would not be visible after one session. We plan to do structured educational sessions that fit the needs of the patients, which is why we set up this study in advance.

The analysis of oral health beliefs is very interesting and well done.

In my opinion, the discussion should be revised by specifying that it is a descriptive study on a small sample, not representative of the French population or of the age group studied 

Thank you for your comment. We have reworded the discussion.

I would prefer that analytical statistics come to validate the assertions of the discussion. 

We provided statistical clarification as mentioned above.

6. PLOS authors have the option to publish the peer review history of their article (what does this mean?). If published, this will include your full peer review and any attached files.

If you choose “no”, your identity will remain anonymous, but your review may still be made public.

Do you want your identity to be public for this peer review? For information about this choice, including consent withdrawal, please see our Privacy Policy.

Reviewer #1: No

Reviewer #2: No

While revising your submission, please upload your figure files to the Preflight Analysis and Conversion Engine (PACE) digital diagnostic tool, https://pacev2.apexcovantage.com/. PACE helps ensure that figures meet PLOS requirements. 

To use PACE, you must first register as a user. Registration is free. Then, login and navigate to the UPLOAD tab, where you will find detailed instructions on how to use the tool. If you encounter any issues or have any questions when using PACE, please email PLOS at figures@plos.org. Please note that Supporting Information files do not need this step.

---

## [Decision Letter · Decision Letter 1]

15 Jul 2021

Which determinants should be considered to reduce social inequalities in paediatric dental care access?  

A cross-sectional study in France.

PONE-D-21-12152R1

Dear Dr. MARQUILLIER,

We’re pleased to inform you that your manuscript has been judged scientifically suitable for publication and will be formally accepted for publication once it meets all outstanding technical requirements.

Kind regards,

Frédéric Denis, Ph.D.

Academic Editor

PLOS ONE

Additional Editor Comments (optional):

Reviewers' comments:

Reviewer's Responses to Questions

**Comments to the Author**

1. If the authors have adequately addressed your comments raised in a previous round of review and you feel that this manuscript is now acceptable for publication, you may indicate that here to bypass the “Comments to the Author” section, enter your conflict of interest statement in the “Confidential to Editor” section, and submit your "Accept" recommendation.

Reviewer #1: All comments have been addressed

Reviewer #2: All comments have been addressed

2. Is the manuscript technically sound, and do the data support the conclusions?

Reviewer #1: Yes

Reviewer #2: Yes

3. Has the statistical analysis been performed appropriately and rigorously? 

Reviewer #1: Yes

Reviewer #2: Yes

4. Have the authors made all data underlying the findings in their manuscript fully available?

Reviewer #1: Yes

Reviewer #2: Yes

5. Is the manuscript presented in an intelligible fashion and written in standard English?

Reviewer #1: Yes

Reviewer #2: Yes

6. Review Comments to the Author

Reviewer #1: (No Response)

Reviewer #2: Dear author,

3 points maybe for next publication.First, it's not because the recruitment of patients is in a teaching hospital that this makes the population studied representative of the general population. 2nd, I maintain that it would have been interesting to specify the zero and once toothbrushing groups for future stratified analysis. Finally, don't underestimate your patients' ability to understand what you are telling them. Contrary to what you indicate in your answer, it is possible that changes in behavior are possible, and therefore that an improvement in hygiene indices would have been observable for a part of the studied population.

7. PLOS authors have the option to publish the peer review history of their article (what does this mean?). If published, this will include your full peer review and any attached files.

Reviewer #1: No

Reviewer #2: No

---

## [Editor Report · Acceptance letter]

23 Jul 2021

PONE-D-21-12152R1 

Which determinants should be considered to reduce social inequalities in paediatric dental care access?
A cross-sectional study in France 

Dear Dr. Marquillier:

I'm pleased to inform you that your manuscript has been deemed suitable for publication in PLOS ONE. Congratulations! Your manuscript is now with our production department. 

Kind regards, 

on behalf of

Dr. Frédéric Denis 

Academic Editor

PLOS ONE